# Isthmin—A Multifaceted Protein Family

**DOI:** 10.3390/cells12010017

**Published:** 2022-12-21

**Authors:** Hosen Md Shakhawat, Zaman Hazrat, Zhongjun Zhou

**Affiliations:** 1School of Biomedical Sciences, The University of Hong Kong, Hong Kong, China; 2The University of Hong Kong Shenzhen Hospital, Shenzhen 518053, China

**Keywords:** isthmin, development, angiogenesis, metabolism, organ homeostasis, immunity, cancer

## Abstract

Isthmin (ISM) is a secreted protein family with two members, namely ISM1 and ISM2, both containing a TSR1 domain followed by an AMOP domain. Its broad expression pattern suggests diverse functions in developmental and physiological processes. Over the past few years, multiple studies have focused on the functional analysis of the ISM protein family in several events, including angiogenesis, metabolism, organ homeostasis, immunity, craniofacial development, and cancer. Even though ISM was identified two decades ago, we are still short of understanding the roles of the ISM protein family in embryonic development and other pathological processes. To address the role of ISM, functional studies have begun but unresolved issues remain. To elucidate the regulatory mechanism of ISM, it is crucial to determine its interactions with other ligands and receptors that lead to the activation of downstream signalling pathways. This review provides a perspective on the gene organization and evolution of the ISM family, their links with developmental and physiological functions, and key questions for the future.

## 1. Gene Organization

Isthmin (ISM) is a secreted protein that was first detected through an unbiased screening for secreted proteins in Xenopus embryos and initially named Xenopus Isthmin (*xIsm*). The ISM protein family has two members, namely ISM1 (~60 kDa) and ISM2 (~63.9 kDa). Both of these proteins contain a hydrophobic signal peptide at the N-terminus along with a centrally positioned thrombospondin type 1 repeat (TSR1) domain. Sequence analysis has revealed that along with the TSR1 domain, ISM also contains another C-terminus domain called the Adhesin-associated domain in MUC4 and other protein (AMOP) domains [1].

## 2. Evolutionary History

The *Ism1* gene has a conserved sequence in various organisms, including humans, chickens, zebrafish, dogs, and cows [2]. *Ism1* is located at human chromosome 20 (20p12.1) and the amino acid (aa) sequence length of 464 is comprised of six exons covering 77.7 kb. In mice, *Ism1* is located at chromosome 2 (2;2F3) and the aa sequence length is 454. In chickens, the location of this gene is at chromosome 3 and the aa sequence is 443, while in zebrafish it is at chromosomal location 13 and the aa sequence is 443 [2]. Phylogenetic analysis demonstrates that both ISM1 and ISM2 proteins are highly conserved in various organisms, thus suggesting the role of the ISM protein family in cellular events. (See Figure 1).

The reliability of each node was estimated by bootstrapping with 1000 replications. The numbers shown at each mode indicate the bootstrap values (%). The alignment was generated by using MUSCLE before constructing the phylogenetic tree and the pairwise deletion method was used for the treatment of missing data. The accession numbers used for Ism1 are indicated as follows: NP_543016.1 (*Homo sapiens*), NP_001012376.1 (*Danio rerio*), NP_001082228.1 (*Xenopus laevis*), **F1NGZ0** (*Gallus gallus*), **F7AJE9** (*Macaca mulatta*), **A0A8C0Q7L4** (*Canis lupus familiaris*), XP_016792946.1 (Pan troglodytes), NP_001157407.1 (Bos taurus), **A2ATD1** (Mus musculus), and **A0A8I5ZXL1** (Rattus norvegicus).

The accession numbers used for Ism2 are indicated as follows: NP_872315.2 (*Homo sapiens*), XP_003314479.1 (Pan troglodytes), XP_003808923.1 (Pan paniscus), **F6RQZ0** (Macaca mulatta), **A0A096NVJ5** (Papio Anubis), XP_019823441.1 (Bos indicus), NP_001018345.1 (Danio rerio), XP_012823996.1 (Xenopus tropicalis), XP_005084068.1 (Mesocricetus auratus), NP_001277231.1 (Mus musculus), XP_008763117.1 (Rattus norvegicus), and **H0XSN9** (Otolemur garnettii).

## 3. TSR1 Domain

The TSR domain is present within the extracellular proteins or in the membrane-bound proteins, and they are associated with multiple functions, including immunity, neuronal development, cell proliferation, cell to cell interaction, cell to matrix interaction, cellular adhesion, migration, and apoptosis [3,4,5,6]. TSR1 contains 60 residues that are comprised of a reverse three-stranded β-sheet and its core is formed by a conserved amino acid residue that includes tryptophan (W), arginine (R), and cysteine (C). A domain-based analysis of the TSR1 domain in ISM1 and ISM2 along with 25 other proteins can be seen in Figure 2.

The accession numbers used for these 27 proteins are mentioned as follows: NP_543016.1 (ISM1), NP_872315.2 (ISM2), NP_922932.2 (ADAMTS6), NP_001269281.1 (ADAMTS10), NP_001311440.1 (ADAMTS12), NP_055087.2 (ADAMTS7), NP_955387.1 (ADAMTS18), NP_620687.2 (ADAMTS16), NP_008969.2 (ADAMTS5), NP_008919.3 (ADAMTS1), NP_001305710.1 (ADAMTS9), NP_079279.3 (ADAMTS20), NP_775733.3 (PAPILIN), NP_620594.1 (ADAMTS13), NP_694957.3 (SBSPON), NP_056019.1 (THSD7A), NP_001033722.1 (RSPO1), NP_001269792.1 (RSPO2), NP_001025042.2 (RSPO4), NP_001138724.1 (PROPERDIN), NP_003957.2 (SEMA5A), NP_001693.2 (ADGRB1), NP_588610.2 (UNC5A), NP_003719.3 (UNC5C), NP_061146.1 (THSD1), NP_116173.2 (RSPO3), and NP_001231818.1 (UNC5B).

The TSR1 domain that forms part of various protein families including spondin, UNC, semaphorins, ADAMTS, brain angiogenesis inhibitor 1 (BAI1), and the human thrombospondin-1 (hTSP-1) protein has a wide range of functions. A previous study found that the TSR domain mediates the binding of F-spondin to the extracellular matrix (ECM) [7], while it is also involved in the binding of SCO-spondin to another extracellular matrix (ECM) named Reissner’s fiber [3], eventually facilitating the removal of mono-amines from cerebrospinal fluid [8]. Furthermore, TSRs are involved in motor neuron and distal tip cell (DTC) migration in UNC-5 as well as axon guidance in the UNC-6 protein, while the presence of multiple TSR domains is liable for diverse functions in various tissues [9]. In the ADAMTS protein family, the TSR domain containing multiple proteins is involved in proteolysis [10], whereas another study has suggested that the TSR domain in the papilin protein mediates inhibition of proteolytic activity in the ADAMTS protein [11]. 

Moreover, anti-angiogenic properties of the TSR domain were observed in the BAI1 protein [12]. hTSP-1 is a glycosylated matricellular protein involved in the regulation of ECM function, whereas the TSR protein interacts with CD36 and negatively regulates angiogenesis, further inducing endothelial cell apoptosis [13,14,15]. TSP1 performs significant roles in tissue repair processes through the TSR domain, which is a recognized regulator in the activation of latent TGF-β and thus in wound healing and fibrosis [16]. 

Two motifs of the TSR domain in TSPI, KRFK and WxxW, are requisite for this latent TGF-β activation. An earlier study demonstrated that the WSXW motif facilitates binding of the TSR domain to the intact heparin [17]. 

The WSLW motif is present in the TSR1 domain of ISM1 and WSPW in ISM2. It is worth investigating whether these two motifs have the same functions, including TGF-β activation. A motif-based analysis of ISM1 and ISM2 can be seen in Figure 3.

Additionally, the CSVTCG motif of the TSR domain facilitates interaction with CD36 that triggers the anti-angiogenic activity via apoptosis of endothelial cells [18,19]. ISM1 also has a CSVTCG motif in the TSR1 domain, and it is also worthy of investigation whether it is also involved in triggering anti-angiogenic function. In ISM1, TSR1 has a DGE motif at the position of 218 to 220, and this is an α2β1 ligand sequence, which is a major collagen receptor on platelets as well as a key co-stimulatory pathway of effector T cells that has been implicated in the pathogenesis of arthritis. The α2β1 integrin is expressed by human Th17 cells and it promotes the survival of effector T cells in humans through the inhibition of Fas-induced apoptosis [20]. Integrin α2β1 facilitated the interaction between embryonic retinal cells and collagen in an avian model [21]. Furthermore, the EPQ motif is present in the TSR1 domain of ISM2, and this motif is known to determine the carbohydrate binding specificity [22]. 

## 4. AMOP Domain

AMOP is only observed in extracellular-domain-containing proteins engaged in cell adhesion, which suggests extracellular localization of the protein. The AMOP domain is ~100 residues long and encompasses eight invariant residues of cysteine, which was initially found to be present in a few splice variants of MUC4 and in four other proteins, Mesh, SUSD2, ISM1, and ISM2 [23] (see Figure 4).

AMOP-containing proteins have conserved cysteine (C) residues, while AMOP in both ISM1 and ISM2 also possesses a KGD motif that binds to the integrin α_IIb_β_3_ present in the multiple antagonists of platelet aggregation and is engaged in the integrin-mediated cellular adhesion and tumour metastasis [24] (see Figure 5).

RGD and KGD are two integrin-bound motifs that were recently identified in the S proteins of SARS-CoV-2 and the ACE2 protein [25,26]. The AMOP domain encompasses an RKD motif that is involved in integrin-dependent cell adhesions [27,28]. However, the RKD motif present in the AMOP domain of ISM1 has been shown to be selective in binding with the extracellular surface of αvβ5 integrins that are engaged in vascular permeability as well as cell migration [29]. Additionally, the AMOP domain in ISM2 also has an WSRL motif that is known to be involved in autophagy induction [30].

## 5. Post-Translational Modifications

Post-translational modifications (PTMs) are considered as backbones of proteins because of their regulatory behaviour over the localization and function of their respective proteins [31]. N-glycosylation is a well-recognized type of post-translational modification in which N-glycans are directly enclosed in Asparagine amino acid residue in proteins, and it is essential for the functional features of proteins, including stability, folding, and secretion [32]. ISM1 is demonstrated as an N-glycosylated secreted protein based on biochemical analysis, and two putative N-glycosylation sites at the positions N39 and N28 were identified, while site-directed mutagenesis by mimicking Asparagine (N) to Glutamine (Q) caused reduction in the ISM1 protein size, thereby indicating that N-glycosylation is required for the secretion of ISM1 which was further validated by tunicamycin treatment [33]. Following that, another study identified two C-mannosylation sites at the TSRI domain, Trp^223^ and Trp^226^, wherein site-directed mutagenesis and tunicamycin treatment demonstrated that C-mannosylation is required for N-glycosylation of ISM1 [34]. Moreover, it has been reported that *C*-mannosylation facilitates protein folding and stabilizing, particularly in the TSR domain [35]. In addition, N-glycosylation in Hyaluronidase 1 (HYAL-1) is required for its secretion and thus regulates enzymatic activity, while upregulation of HYAL-1 is reported to be linked to tumour cell proliferation and angiogenesis in multiple cancers [36,37,38]. Moreover, N-glycosylation of ADAM8 and ADAM13 proteins is crucial for its functional features, including processing, stability, and activity [39,40]. 

## 6. Expression and Function

ISM1 is expressed in a large number of human tissues, such as lung, liver, breast, brain, stomach, muscle, skin, bone marrow, and colon. Spatiotemporal analysis indicated that in mouse embryos, *Ism1* transcripts were observed in the anterior mesendoderm (AM), paraxial and lateral plate mesoderm (LPM), mid-brain hindbrain boundary (MHB), and trunk neural tube, while in adult stages, the highest expression level of *Ism1* was found in the lung and the brain and a moderate expression level was observed in the heart, kidney, ovary, testis, and bone marrow [2]. On the other hand, a different expression pattern was observed in chicken embryos, such as that vigorous expression of *Ism1* transcripts was observed in the anterior head mesoderm, optic vesicles, LPM, trunk neural tube, and dermomyotome, whereas a strong expression level was observed in the ear, eye, and spinal cord primordia in later stages [2]. Moreover, another previous study identified *ism1* amongst the top 20 genes which are expressed in the site of anterior primitive streak (APS) in the chicken embryo [41]. Interestingly, asymmetric expression of *Ism1* led to the study on the embryonic role of ISM1 on chicken embryos, whereas defective NODAL signalling in the left LPM as well as impaired asymmetric heart morphogenesis were triggered by ISM1, and thus, it was identified as an antagonist of NODAL signalling which plays role in asymmetric organ morphogenesis [33]. This study also revealed that through the C-terminus AMOP domain, ISM1 negatively regulates NODAL signalling followed by suppressing activation of effector molecule SMAD2 [33]. 

Additionally, ISM1 is expressed since the initial stages of embryonic development in both Xenopus and zebrafish. *Ism1* has strong maternal expression in Xenopus eggs and other expression domains in the embryo, including the blastopore lip, paraxial mesoderm, neural folds, cranial neural crest, and ear placode. While the expression pattern is similar to fgf8, ISM1 was demonstrated as part of the FGF8 synexpression group, including the *Spry* and *Sef* genes [1]. 

In addition, ISM1 expression was also found in the brain, heart, eye, and spleen tissues of zebrafish. In zebrafish embryonic development, *Ism1* was identified at very early developmental stages, including the tail bud stage, shield stage, and early somitogenesis, while the expression of *Ism2* could not be identified up to 24 h post fertilization (hpf). This is despite the fact that *Ism2* expression was most vigorous in two bilateral streams of the mesenchymal cells in the head region and moderate expression was found in trunk [42]. Furthermore, *Ism1* knockdown in zebrafish embryos gave rise to the aberrant formation of intersegmental vessel (ISV) in the trunk [28]. Previously, upregulation of *Ism1* in the zebrafish was identified in relation to the WNT/β-catenin signalling pathway [43], and in another study, *Ism1* was also identified as a NODAL-controlled gene [44]. Additionally, *Ism1* expression in blastoderm required nodal signalling and its subsequent expression, even though the authors observed a compensated function in embryonic development [42] which is opposite to that observed in chicken embryos. Spatiotemporal expression of *Ism1* and being a target gene of multiple signalling pathways as mentioned above indicates there might be other functions of ISM in multiple biological processes. Over the past few years, multiple studies have focused on the functional analysis of ISM1 in several events, including angiogenesis, metabolism, organ homeostasis, immunity, craniofacial development, and cancer. 

## 7. Angiogenesis

Angiogenesis is a tightly controlled process of forming new capillary blood vessels from pre-existing ones which generates oxygen and nutrients for cells [45]. Following birth, angiogenesis continues to facilitate organ growth; nevertheless, in adulthood, blood vessels mainly persist quiescently, while angiogenesis only occurs in multiple physiological circumstances, such as wound healing, damaged tissue repair, embryonic development, and organogenesis [46]. A plethora of growth factors, cytokines, and secreted proteins participate in the angiogenesis process [47,48]. In addition, angiogenesis could be inhibited by endogenous inhibitors, which are proteins formed and secreted in the body that may inhibit blood vessel formation.

A previous study has demonstrated ISM1 as a novel inhibitor of angiogenesis [49], whereas recombinant mouse ISM1 (rISM1) was expressed in *E. coli* and used to treat human umbilical vein endothelial cells (ECs) in vitro. Here, it was found that through the AMOP domain, ISM1 can inhibit capillary network formation in the initial stages of angiogenesis via its interaction with αVβ5 integrin without affecting EC migration. Additionally, rISM1 is reported to inhibit vascular endothelial growth factor (VEGF)-mediated angiogenesis, while overexpression of ISM1 suppressed tumour angiogenesis in B16 melanoma cells and tumour growth in mice [49]. 

Furthermore, integrins are cell-surface-signalling receptors involved in the activation of multiple intra-cellular signalling pathways as well as mediation of adhesion to the ECM and adjacent cells [50]. Additionally, integrins also serve as regulators of multiple processes, including cell growth, proliferation, migration, differentiation, apoptosis, and tissue repair [51]. Integrins are also reported to be instantly inhibited by means of antibodies, peptides, and peptidomimetics [52]. 

Immobilized ISM1 in the ECM promotes EC survival, while conversely, soluble ISM1 induces EC apoptosis via αvβ5 integrin as an antagonist and indicates that the anti-angiogenic functions of ISM1 are diminished when it is present in immobilized form [28]. Contrary to ISM1, other signalling molecules including vitronectin and fibronectin serving as agonists do not exhibit dual function based on their soluble or insoluble state. 

Glioma is a common central nervous system (CNS) tumour that accounts for 30% of all intracranial tumour incidences, and angiogenesis is vital to the formation of malignant gliomas. Furthermore, ECM molecules promote cell signalling via activation of particular cell adhesion receptors, access modulation to soluble factors, and modification of mechanical properties of the tissue (Hynes, 2009). Additionally, an earlier study [53] investigated the role of ISM1 in glioma angiogenesis and found that it can hinder HUVEC proliferation by the inhibition of VEGF. Furthermore, it was also demonstrated that ISM1 activated the HUVEC apoptosis through the caspase-3 pathway, and thus, inhibits tumour angiogenesis in vivo [53]. 

At present, the role of ISM2 in angiogenesis still remains elusive, while it is not surprising that it also contains the TSR1 domain, which has been observed in multiple anti-angiogenic proteins. For instance, the anti-angiogenic function of ADAMTS1 requires the TSR1 domain [54], while it has anti-angiogenic activity in primary gastric cancer [55]. Furthermore, ADAMTS2 is reported to inhibit the formation of the capillary network in EC 3D culture models [56], whereas ADAMTS4 is also reported to suppress angiogenesis and melanoma growth [57]. Additionally, ADAMTS5 was also reported to inhibit angiogenesis through the TSR1 domain [58]. Moreover, SCO-spondin (SCOP) is also reported to function as an angiogenesis inhibitor in glioblastoma [59], while the UNC5B protein is involved in angiogenesis inhibition [60,61]. In summary, TSR1-domain-containing proteins seem to show anti-angiogenic functions, while according to our TSR1 domain sequence alignment analysis, ISM2 has a WSPW motif that is found to be conserved in ADAMTS12, SBSPON (somatomedin B and TSP-1-domain-containing protein), RSPO1 (R-spondin-1), and SEMA5A. It is noteworthy that this heparin-binding motif WSPW in the TSR domain was reported to mediate the inhibition of angiogenesis of ECs [62] and anti-proliferative activity of TSPs [63]. The participation of ISM1 and the above-mentioned TSR1 domain containing secreted and ECM proteins in anti-angiogenesis suggests that this domain might be considered for the development of novel anti-angiogenesis therapeutic strategies. 

## 8. Pathological Processes

In the studies listed above, ISM1 has been demonstrated as an endogenous angiogenesis inhibitor, and it might be interesting to elucidate whether this protein inhibits cancer to generate its own blood vessels. However, over the past few years, increasing evidence has revealed that the atypical expression of ISM1 may affect cancer. 

Long non-coding RNAs (lncRNAs) perform a pivotal role in the progression and metastasis of a variety of carcinomas, while lncRNA H19 (H19) is highly prevalent in gastric cancer tissues [64]. Another study demonstrated that ISM1 is a binding protein of lncRNA H19, which mediates ISM1 upregulation and boosts carcinogenesis and metastasis of gastric cancer (Li et al., 2014).

Furthermore, in hepatocellular carcinoma (HCC), cell proliferation and migration are reported to be promoted by ISM1 and regulated by the interaction with circular RNA (circRNA) and micro RNA (miRNA), namely hsa_circ_0091570/miR-1307 hsa_circ_0091570, which competitively binds to miR-1307 by serving as the ceRNA (competing endogenous RNA) that then regulates ISM1 expression [65]. This study summarized that downregulation of hsa_circ_0091570 occurs in HCC and could act as ceRNA through sponging miR-1307 for regulating ISM1 expression and thereby taking part in HCC progression [65]. 

Additionally, colon adenocarcinoma (COAD) is a malignant tumour of the digestive tract that is associated with an extremely high incidence rate and is ranked third amongst all tumours globally, and miRNA is reported to play a major role in this tumour cell proliferation and apoptosis [66,67]. The Wnt/β-catenin signalling pathway serves as a regulatory pathway in tumorigenesis because of its involvement in multiple cellular processes [68], while Wnt3a is substantially enhanced in colon cancer cells in promoting tumour angiogenesis and metastasis [69]. A recent study [70] has shown that miR-1307–3p inhibits the activation of the Wnt3a/β-catenin signalling pathway by targeting and thus downregulation of ISM1, which then inhibits proliferation and promotes the apoptosis of COAD cells. Conversely, ISM1 overexpression promoted activation of the Wnt3a/β-catenin signalling pathway along with proliferation and decelerated cell apoptosis in COAD cells. 

Furthermore, a recent study has also reported the elevated expression of ISM1 in the CRC tissue of patients, and on top of that, a positive correlation of ISM1 with cancer-associated signalling pathways including EMT, hypoxia, KRAS, Notch, and Hedgehog were observed [71].

The inner surface of the micro vessels is lined by endothelial cells to generate a semi-permeable barrier that allows exchange of fluids and proteins in blood and tissue, while tight regulation of this endothelial permeability is crucial in the maintenance of organ homeostasis, and thus, dysfunction including endothelial hyperpermeability leads to vascular inflammation related to multiple diseases, such as respiratory distress, acute lung injury, sepsis, diabetes, and cancer [72,73]. Identification of functional signalling molecules involved in vascular permeability (VP) for circulatory homeostasis might have therapeutic potential. ISM1 is demonstrated as a novel VP inducer which acts via cell-surface GRP78-facilitated activation of Src followed by Src-mediated tyrosine phosphorylation of adherens-junction (AJ) proteins and the consequent dissociation of these AJ proteins, therefore resulting in barrier disruption [74]. Furthermore, this study [74] also reported that a lipopolysaccharide (LPS)-induced acute lung injury (ALI) mouse model had elevated upregulation associated to ALI and thus suggested the inhibition of ISM1 overexpression as a therapeutic strategy in ALI.

On the top of that, hypoxia is one of the most common reasons for vascular hyperpermeability and a crucial risk factor of several pathological characteristics in lung disorders [75]. Hyperpermeability of pulmonary microvascular endothelial cell (PMVEC) monolayers induced by hypoxia is vital for vascular leakage and leads to pulmonary diseases including ALI and high-altitude pulmonary oedema (HAPE). AECII are cuboidal cells that comprise about 15% of the overall lung cells, and they account for epithelium reparation and facilitate lung homeostasis through the secretion of several lysozymes and proteins. The elevated ISM1 level in AECII was accountable for hypoxia triggered PMVEC monolayer hyperpermeability in an AECII/PMVEC co-culture system that indicated substantial function of alveolar epithelial cells and a modulatory role of ISM1 in hyperpermeability featured lung diseases [76]. Hypoxia-inducible factor-1α (HIF1α) is a master transcriptional regulator of hypoxia, while the same study demonstrated that elevated HIF1α expression transcriptionally activated *Ism1* gene expression and thus identified *Ism1* as a novel HIF1α target gene [76]. 

It is already clear that inflammation control and maintenance of homeostasis are of utmost importance for lung health, whereas prolonged lung inflammation leads to chronic obstructive pulmonary disease (COPD), and the severity of disease in COPD patients is directly linked to the accumulation of alveolar macrophages (AMs). It is important to highlight that, by using genetic (*Ism1*) and pathological (COPD) mouse models, one study reported that the secreted protein ISM1 is lung-resident, having a high expression that safeguards lung homeostasis through controlling AM numbers and an efficient phenotype through cell-surface GRP78 (csGRP78)-facilitated apoptosis [77]. Furthermore, the same study revealed that intratracheal delivery of recombinant ISM1 (rISM1) exhibited effective suppression of lung inflammation through the depletion of the pro-inflammatory cs-GRP78^high^ AMs by targeted apoptosis, as well as prevented emphysema development and thus retained pulmonary function in cigarette-smoke-induced COPD mice. Furthermore, *Ism1* knockout in mice exhibited an increase in cs-GRP78^high^ AMs along with upregulation of MMP9, MMP12, and NF-κB p65, in addition to a moderate increase in TGF-β1 and VEGF-A, prolonged lung inflammation, and progressive emphysema [77]. 

On the other hand, GRP78 is a stress-response protein that belongs to the heat-shock protein family, which has the ability to modulate protein folding and is upregulated in cells under stress. This protein is known to be highly expressed in various human cancers, including melanoma and breast, prostate, lung, and ovarian cancer, and thus it is associated with chemoresistance, increased malignancy, and inadequate patient outcomes [78,79]. Of note is that ISM1 selectively promotes cellular apoptosis, harbouring elevated cell-surface GRP78 in activated ECs as well as in metastatic and aggressive cancer cells, while systemic delivery of the GRP78-specific cyclic peptide BC71 effectively suppressed the growth of subcutaneous tumours in mice [79]. 

Moreover, orofacial clefts have a complex aetiology and are one of the most prominent birth defects, affecting 1–2 children per 1000 births. Complex diseases have been reported to be linked with copy number variants (CNVs) and ISM1 heterozygous deletions were shown to be enriched in cleft lip and palate cases compared to the controls, therefore indicating that the loss of even one copy predisposes to this disorder (Lansdon, Darbro, Petrin, Hulstrand, Standley, Brouillette, Long, Mansilla, Cornell, Murray, et al., 2018). SM1 is expressed in the same synexpression group as other clefting genes, including spry1, spry2, and fgf8, and knockdown of ISM1 causes craniofacial dysmorphologies in frogs, including cleft-like phenotypes, hence confirming the role of ISM1 in craniofacial development (Lansdon, Darbro, Petrin, Hulstrand, Standley, Brouillette, Long, Mansilla, Cornell, Murray, et al., 2018).

Additionally, trophoblastic cells in placental tissues collected from patients with gestational hypertension and preeclampsia have a strong expression of ISM1, while reduced serum concentrations of ISM2 were observed in preeclampsia patients and, quite the opposite, ISM2 showed prominent expression in choriocarcinoma, thus suggesting a possible contrast in function [80]. Furthermore, strong expression of ISM2 is observed in choriocarcinoma, while moderate expression in lung and prostate adenocarcinoma is observed and mild expression is indicated in colon adenocarcinoma and cohesive gastric carcinoma [80]. 

In summary, ISM1 upregulation and overexpression are reported in multiple cancers, including gastric cancer, hepatocellular carcinoma (HCC), colon adenocarcinoma, and colorectal cancer, while ISM2 overexpression was reported in choriocarcinoma. However, multiple signalling molecules and pathways are also reported as being involved in the regulation and synexpression. Even though ISM1 is reported as an endogenous angiogenesis inhibitor, it is quite the opposite that overexpression of ISM1 and ISM2 is found in the above-mentioned cancers, while it could be interesting to explore whether tight regulation over the tissue-specific expression of these proteins might play a role in cancer therapy. 

## 9. Metabolism

Endocrine tissues have enhanced expression of ISM1, including the thyroid, pituitary, and adrenal glands, whereas the thyroid gland is reported to have the highest expression of this protein. Furthermore, as discussed earlier, increased expression of ISM1 is found in adipose tissue, pancreas, spleen, liver, and kidney compared to other organs. Secreted by adipose tissue, adipokines participate in systemic metabolism and tissue homeostasis in a paracrine or endocrine manner, while imbalance in adipokines causes metabolic dysfunction. A recent study [81] reported ISM1 as an adipokine secreted by mouse adipocytes that promotes glucose uptake by a signalling cascade which involves an unidentified receptor kinase such as insulin. Furthermore, recombinant ISM1 led to a vigorous rise in GLUT4-dependent glucose uptake in both murine and human adipocytes as well as primary human muscle cells, whereas *Ism1* knockout in adipocytes exhibited a reduction in glucose uptake and insulin-dependent phosphorylation of protein kinase AKT at the serine residue 473 [81].

Furthermore, complete knockout of *Ism1* in mice exhibited a drop in glucose tolerance as well as a decline in glucose uptake in the brown adipose tissue and muscle, while on the other hand, overexpression of *Ism1* in diet-induced mice led to the reduction in adipose tissue mass along with improved glucose tolerance, insulin sensitivity, and boosted suppression of hepatic glucose production in an insulin clamp [81]. 

On the other hand, even though mouse adipocytes and hepatocytes exhibited ISM1-dependent signalling cascade associated improvised glucose uptake, astonishingly ISM1 suppressed insulin-dependent de novo lipogenesis (DNL). In addition, it was suggested that ISM1 facilitated hepatic DNL inhibition, which prevented lipid accumulation in an NAFLD mouse model, and thus the authors argued that ISM1 signalling indirectly modulates the hepatic lipid accumulation by inhibiting the fatty acid releases from the adipose tissue [81]. 

Furthermore, a study conducted on Spanish pubertal boys and girls revealed higher ISM1 levels in obese pubertal boys as compared to normal weight and overweight boys, while there were no significant changes observed in pubertal girls, thus indicating a plausible function of ISM1 in male restricted obesity [82]. 

However, a cohort study conducted on Canadian young adults with childhood obesity history revealed a positive correlation of obesity with risks of T2DM and coronary artery disorders [83]. In contrast and as previously mentioned, placental tissues from patients with gestational hypertension and preeclampsia are associated with a strong expression of ISM1 [80]. However, a cohort study conducted on women with gestational diabetes mellitus (GDM) demonstrated that women who developed preeclampsia did not show more insulin resistance as compared to non-preeclamptic women [84]. However, ISM1 function in glucose homeostasis is enigmatic, no receptor has been identified so far for this adipokine in functional studies, and there is also an absence of studies reporting on ISM2 and its role in metabolism and any other related processes. 

## 10. Immune System

Haematopoiesis is the formation of blood cells in the marrow that involves the replication as well as specialization of hematopoietic stem cells (HSCs) towards the downstream progenitor cells [85]. Multiple classes of secreted proteins, among them the colony stimulating factors (CSFs) or haematopoiesis, are known to function as important regulators during hematopoietic differentiation. Furthermore, these secreted molecules are of clinical significance because of their application in stimulating haematopoiesis in patients with neutropenia and several other haematological diseases [86]. Consequently, identifying the function of novel secreted proteins that show specific spatiotemporal expression patterns within the hematopoietic cells is of considerable interest. ISM1 is reported as required for the normal formation of HSPCs towards their downstream progeny during haematopoiesis in zebrafish because *Ism1* knockout resulted in the drop in frequency of mature blood cell populations, including neutrophil, macrophage, and erythrocyte, while elevated expression of *Ism1* was reported in stromal cell lines [87]. However, the study was conducted on a zebrafish morpholino-based model, and it remains unclear how ISM1 interacts with HSPCs and the underlying mechanism requiring ISM1 during haematopoiesis in zebrafish [87]. 

Moreover, ISM1 was found to be produced in human and mice barrier tissues, such as mucosa, skin, and some lung lymphocytes, which might be linked to the NK, NKT, and Th17 cell lineages. In addition, the ISM1 expression increases significantly in CD4+ T cells when the cells are polarized to the Th17 lineage in vitro, indicating that ISM1 is a mediator of lymphocyte effector functions and might be involved in both innate and adaptive immune responses [88].

On top of that, a recent study found a role of ISM1 in immunity against viral infections, as its expression was induced by Grass carp reovirus (GCRV) both in vitro and in vivo, while they further demonstrated that rISM1 reduced the cytopathic effects in GRCV-infected cells and promoted the *Ifn* gene and IFN-inducible antiviral protein *Mxa* gene [89]. Additionally, the same study proposed that ISM1 may first induce the activation of the Tbk1–Irf3–Ifn antiviral signalling pathway, which then leads to enhanced expression of *Ifn* and *Mxa*, both of which in turn suppress viral replication [89]. 

Additionally, as discussed earlier, a study demonstrating ISM1 upregulation in CRC patients also revealed the associations between suppressive immune cells (M2 macrophages, T-regs, and T cell exhaustion), and ISM1 overexpression detected in PD1-resistant patients further indicated that ISM1 upregulation can possibly play a crucial role in the formation of an inhibitory immune microenvironment [71].

## 11. Mechanism and Receptors

Secreted proteins can play a direct role in the regulation of biological processes, including cell proliferation, development, adhesion, migration, and apoptosis [90]. As from the above-mentioned studies and discussions, it is clear that ISM1 is involved in multiple biological processes.

However, since ISM1 is a circulating ligand, it is crucial to elucidate its downstream tissue-specific functions mechanistically. In an earlier study [49], the authors reported that ISM1 can bind to *α*v*β*5, although they did not demonstrate the signalling properties of ISM1, while further suggesting that ISM1 can function either as an antagonist or agonist which modulates EC apoptosis or survival depending on its physical state. 

A previous study conducted by our lab revealed that ISM1 regulates the NODAL signalling by binding to activin receptor type IB during chicken foetal development and is involved in organ asymmetry. Furthermore, it is noteworthy that ISM1 has minimal effect on ACTIVIN-A, BMP4, or TGF-β1 signalling but significantly inhibits NODAL-induced phosphorylation of SMAD2 [33]. 

In addition, a previous study reported that ISM1 activates the Src pathway through the novel ISM-GRP78 apoptosis pathway, triggering apoptosis not only in endothelial cells but also in in cancer cells with an elevated expression of cell-surface GRP78 [91]. 

Moreover, a critical role of ISM1-csGRP78 signalling is reported in lung homeostasis by controlling alveolar macrophage (AM) number and function, whereas the association between ISM1 and COPD pathogenesis in mice was observed as rISM1 delivery suppressed inflammation, attenuated emphysema, and preserved lung function by specifically targeting csGRP78 on stress-activated csGRP78^high^ AMs in CS-induced COPD mice [77]. Furthermore, ISM1 is also reported to promote vascular permeability through cell-surface GRP78-mediated Src activation [74]. Additionally, ISM1 is reported to promote the Tbk1–Irf3–Ifn antiviral signalling pathway, which then leads to enhanced expression of *Ifn* and *Mxa*, both of which in turn suppress viral replication [89]. 

Of note, it was [81] reported that ISM1, as an adipokine secreted by mouse adipocytes, promotes glucose uptake by induction of the PI3K-AKT pathway, which requires mTORC2.

In zebrafish, ISM1 upregulation of ISM1 was observed to be linked to the overexpression of Wnt signalling during embryonic development [43]. It is noteworthy that *Ism1* synexpression with FGF8 was observed in Xenopus embryos [1], while *Ism1* was identified as a NODAL controlled gene [44]. 

Furthermore, Wnt3a is substantially enhanced in colon cancer cells in promoting tumour angiogenesis and metastasis [69]. A previous study [70] has shown that miR-1307–3p inhibits the activation of the Wnt3a/β-catenin signalling pathway via targeting ISM1 downregulation and thus inhibiting proliferation and promoting the apoptosis of COAD cells. Conversely, ISM1 overexpression promoted the activation of the Wnt3a/β-catenin signalling pathway along with the proliferation and decelerated cell apoptosis in COAD cells. Furthermore, another study has also reported the elevated expression of ISM1 in the CRC tissue of patients, and on the top of that, the positive correlation of ISM1 with cancer-associated signalling pathways including EMT, hypoxia, KRAS, Notch, and Hedgehog was observed [71].

Transforming growth factor-β (TGF-β) is a conserved family of secreted polypeptide factors which regulate several aspects of physiological embryogenesis and adult tissue homeostasis [92]. This family includes many important proteins, such as TGF-β, nodal, lefty, activin, growth and differentiation factor, and bone morphogenetic proteins [93]. Taking into account previous findings about the relationship of ISM1 with the nodal [33] and TGF-β [77] pathways and the importance of these pathways in several crucial processes, it is important to further explore and elucidate ISM1’s role in the TGF-β pathway, particularly focusing on the interaction between ISM1 and TGF-β receptors.

## 12. Future Perspectives

Even though ISM was identified two decades ago, there remains a lack of understanding with regards to the roles of ISM1 and ISM2 in embryonic development and other physiological processes. To address the role of ISM, functional studies have begun but unresolved issues for researchers remain. Initially, its role in development [29,33,42] and physiology [77,81,82,89] has been addressed but still needs further detailed studies.

The truth is, although multiple studies have identified the spatiotemporal expression of ISM during embryonic development of chickens, mice, zebrafish, and Xenopus, the exact roles and signalling cascades working behind the expression are yet to be discovered, except that one study revealed that ISM1 negatively regulates NODAL signalling in asymmetric organ morphogenesis of chicken embryos [33]. As the functional studies of ISM have been studied in embryonic development, it is possible that several human birth defects and genetic diseases will be attributed to the mutations of the *Ism* gene family. 

To elucidate the regulatory mechanism of ISM, it is crucial to determine its interactions with other ligands and receptors that lead to the activation of downstream signalling pathways. Even though ISM1 is reported to bind with GRP78 [74,77,91] and αVβ5 [28], its diffusion into the ECM to activate specific receptors remains elusive. 

Another major frontier will be to elucidate its role in angiogenesis and cancer. ISM1 was demonstrated as an endogenous inhibitor of angiogenesis by two studies [49,53], while one study demonstrated that the solubility of the protein itself determines its angiogenic properties [28]. However, both in cells and in intact tissues, receptor-based studies are needed to establish it as an endogenous inhibitor of angiogenesis. Endogenous angiogenesis inhibitors affect the pathological angiogenesis process without affecting quiescent capillary vessels for tissue homeostasis, and thus, based on that, the development of novel anti-angiogenic proteins is needed along with determining the ways to regulate the expression by identification of associated signalling molecules. This can be carried out by generating knockout mice models, particularly tissue specific knock-out studies can be more significant.

Even though ISM1 is reported as anti-angiogenic, multiple studies have reported the prominent expression of ISM1 in several cancers and the elevated expression of ISM2 in choriocarcinoma. At this point, the function of these proteins remains enigmatic, and thus there is a need to explore the bioactivity of ISM as well as the regulatory mechanism involved in the secretion of the protein at both the cellular and tissue level. Additionally, future studies are required to find whether ISM activation itself is an aetiological agent in initiating the primary tumour or a progression factor in cancer pathogenesis. These studies may further lead to the development of pharmacogenetic agents to treat various diseases and cancers. 

Recently, Lam et al. [77] found that ISM1 possesses therapeutic potential in vivo, as its administration alleviated pathological conditions in a COPD mouse model. Moreover, dietary intake of rISM1 in *N. guentheri* was shown to prolong the mean lifespan by 7.5% [94]. Future investigations should elucidate the appropriate method of delivery, its long-term consequences, and its optimal dosage level for therapeutic applications. 

## Figures and Tables

**Figure 1 cells-12-00017-f001:**
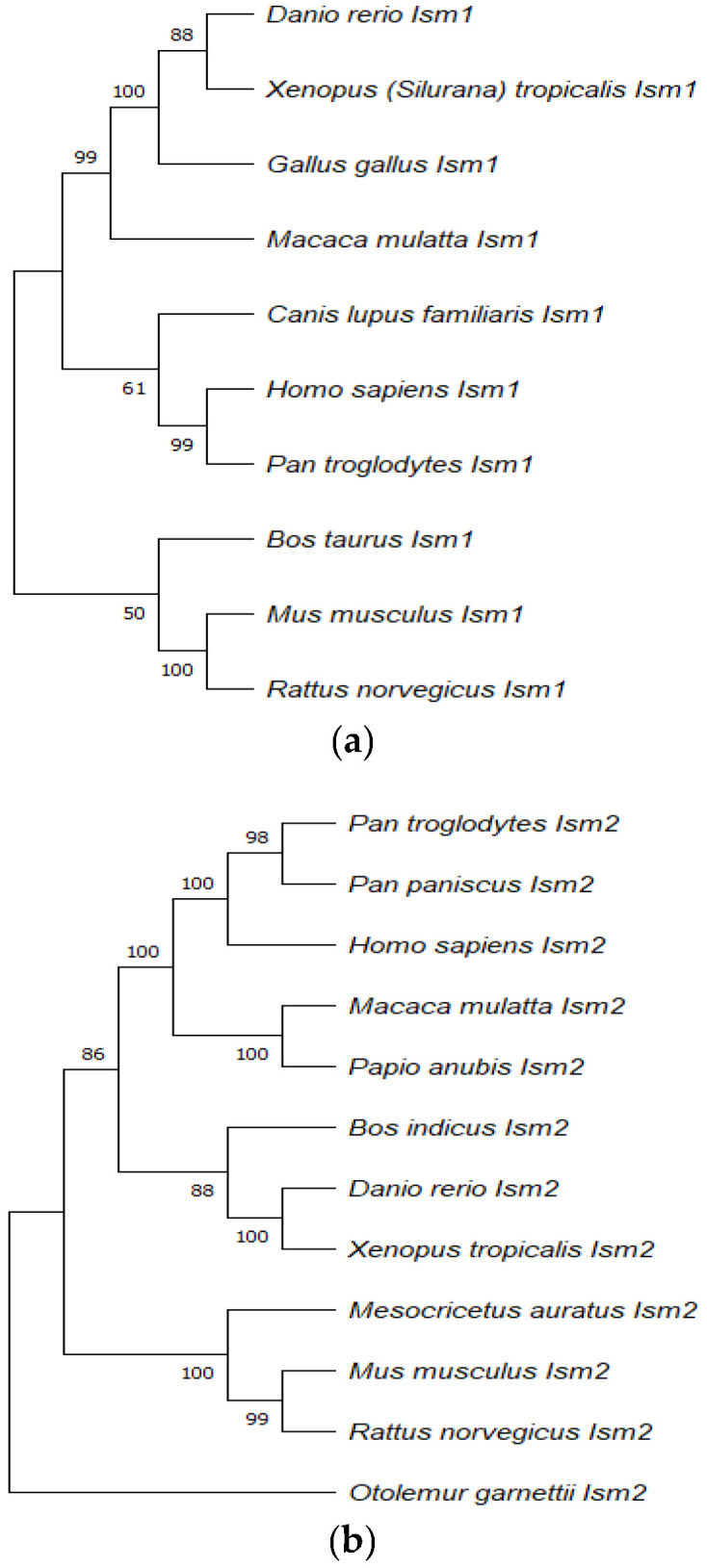
The phylogenetic tree of ISM1 (**a**) and ISM2 (**b**) proteins were drawn by Molecular Evolutionary Genetic Analysis (MEGA 11.0) using the neighbour-joining method.

**Figure 2 cells-12-00017-f002:**
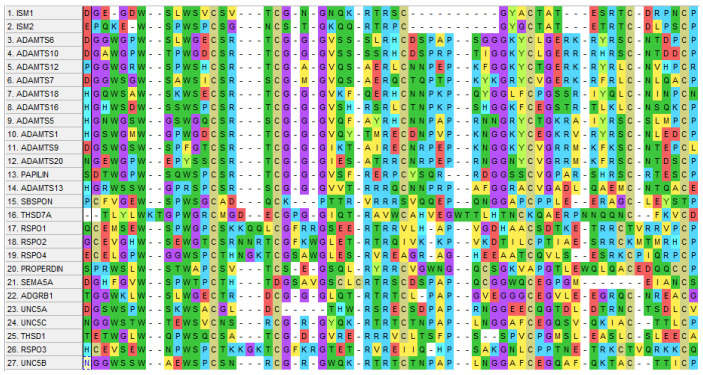
Schematic representation of human TSR1 domain sequence alignment of 27 proteins drawn by Molecular Evolutionary Genetic Analysis (MEGA 11.0) using MUSCLE alignment.

**Figure 3 cells-12-00017-f003:**
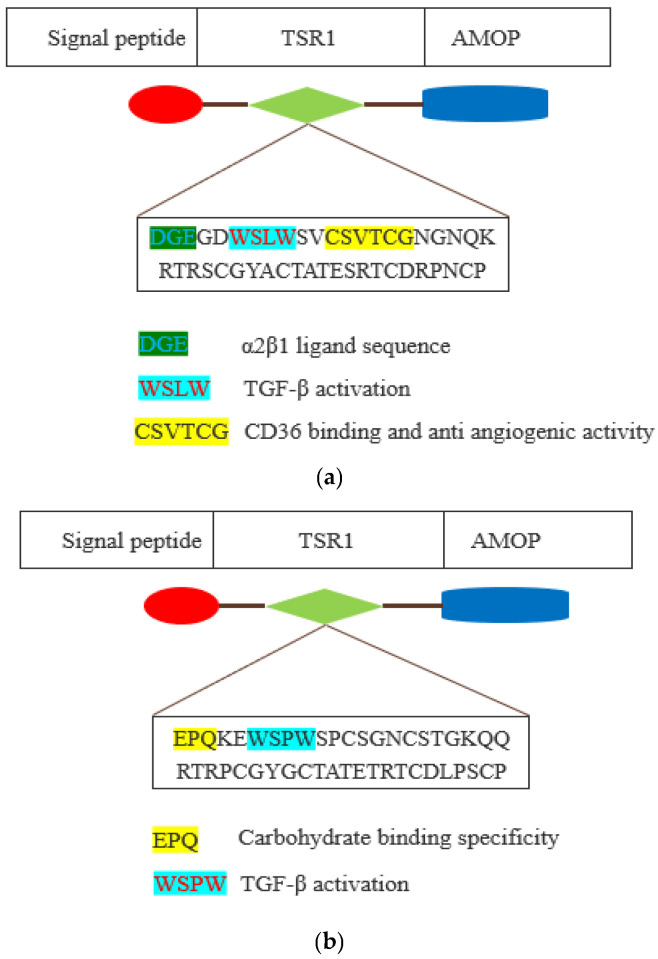
Schematic representation of thrombospondin type 1 repeat (TSR1) domain of ISM1 (**a**) **and ISM2** (**b**). The TSR1 domain is depicted by green-coloured shape. The amino acid sequence of TSR1 domain motifs is shown as coloured with their existing and plausible functions.

**Figure 4 cells-12-00017-f004:**
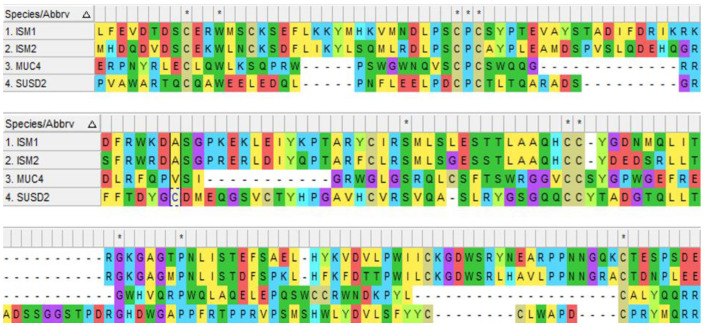
Schematic representation of human AMOP domain sequence alignment in 4 proteins drawn by Molecular Evolutionary Genetic Analysis (MEGA 11.0) using MUSCLE alignment. The accession numbers used for these 4 proteins are mentioned as follows: NP_543016.1 (ISM1), NP_872315.2 (ISM2), NP_004523.3 (MUC4), and NP_062547.1 (SUSD2).

**Figure 5 cells-12-00017-f005:**
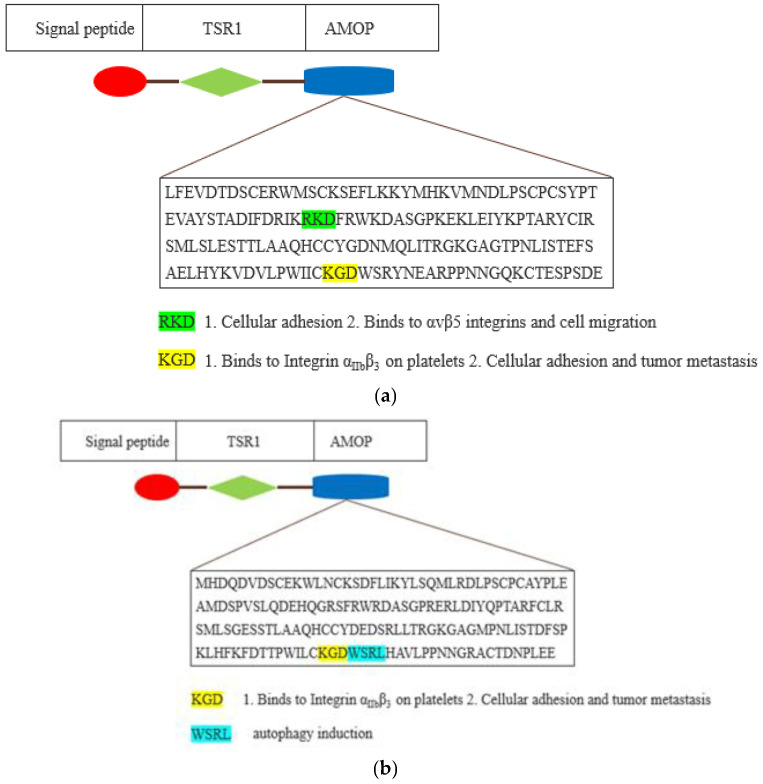
Schematic representation of AMOP domain of ISM1 (**a**) and ISM2 (**b**). The AMOP domain is depicted by blue-coloured shape. The amino acid sequence of AMOP domain motifs is shown as coloured with their existing and plausible function.

## Data Availability

Not applicable.

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
