# Peer review of "Isthmin—A Multifaceted Protein Family"

_cells, 2022, doi:10.3390/cells12010017_

Round 1

Reviewer 1 Report

The manuscript entitled, “Isthmin – A multifaceted protein family,” by Shakhawat and colleagues provides a review of the enigmatic ISTHMIN genes, particularly focusing on ISTHMIN 1. ISTHMIN 1 has been the focus of a large number of studies, yet its role in development and disease is still unclear. This manuscript summarizes many of these disparate studies with the goal of providing insight into ISTHMIN function. While the review does a good job in coalescing a large amount of published data, the overall organization needs to be improved, and important published results have been left out. Finally, a large number of grammatical errors are present throughout the manuscript, which needs to be overhauled by careful attention to wording and sentence structure. In summary, there are a number of significant concerns that need to be addressed, and these are listed below:

1.  In the abstract, lines 12 and 13, craniofacial development should be included as one of the several events.

2. Lines 33-26: “Ism” should be replaced with “Ism1” since the genomic locations are all indicated for Ism1 and not Ism2. Along the same lines (but a more minor concern), why not report the locations of Ism2?  

3. Line 39: “Mentioned” should be replaced with “indicated.” This is one example of the grammatical issues throughout.

4. Following line 58: There is no summary of what Figure 1 is meant to show. Some sort of statement should be made regarding the level of ISM1 conservation, such as, “these data indicate that ISM1 is a highly conserved protein, suggesting that it plays important roles in the cell….” Also, it should be made clear that the phylogenetic analysis was for proteins and not genes (this could be put in the figure legend).

5. Figure 2 and its legend should follow line 66 (just after “….and cysteine (C). This makes more sense for the organization of the review, since it’s best to put the graphic before describing all the details regarding the TSR domain.

6. Similar to number 5, the line (102) “Motif based analysis of ISM1 and ISM2 can be seen in Figure 3” would be best if inserted directly after the line (106) “WSLW motif is present in the TSR1 domain of ISM1 and WSPW in ISM2” since Figure 3 annotates precisely these regions.

7. For Figure 3, there is no a) or b) for the two halves of the cartoon.

8. Figure 4: The alignment should start with ISM1 and then ISM2 followed by the other proteins. This way, readers can better appreciate the level of conservation between the two ISM proteins.

9. Figure 5: Once again, there is no a) or b) for the two halves of the cartoon.

10. Line 181: “ISM is expressed in several human tissues” should be changed to “ISM1 is expressed in a large number of human tissues,” and it would be good to include additional examples such as muscle and skin. See Genecards for a nice compendium of tissues analyzed.

11. Lines 220-221: It is unclear what is meant by the statement, “this may be because of sequence variation between the zebrafish and chick in downstream of signal peptides.” Please rewrite.

12. Line 225: “craniofacial development” should be included in the list of events.

13. ISM1 has been shown to interact with integrins. However, this review does not address all the functions of integrins (other than a few functions as in lines 245-248), which include adhesion to the ECM, proliferation, apoptosis, cell migration, differentiation, and others. Many of these functions overlap with what has been seen for ISM1 knockdown or expression studies. Thus, a brief section detailing all the various functions of integrins is warranted, which in turn will allow readers to recognize how ISM1 may function through integrins. Along these lines, one key function of integrins that may tie many of the ISM1-related processes described in the review together is cell migration, since migration is a key component of any cells that have to move from one place to another (such as lymphocytes).

14. The Lansdon et al paper (2018, which is improperly cited in the References section with regard to authorships) should be summarized in the “Pathological processes” or comparable section, as it is the only study that strongly links a human congenital anomaly (birth defect) with ISM1 copy number reduction (this is especially important, as the authors mention that “it is possible that several human birth defects and genetic diseases will be attributed to the mutations of Ism1 gene family” (lines 541-543) yet do not report the deletion of ISM1 association with a birth defect from this published study). The following points should be considered when writing the summary:

a. Heterozygous deletions of ISM1 were found to be enriched in cleft lip and palate cases compared to controls, and ISM1 scores as a likely haploinsufficiency locus (meaning loss of one copy predisposes cases to a disease/disorder).

b. Knockdown of ism1 in frogs produced cleft-like phenotypes in frogs, and ism1 is strongly expressed in the cranial neural crest cells of the branchial arches during face development including BA1 which patterns the primary and secondary palates (these structures fail to properly develop in cleft lip and palate cases).

c. ism1 is expressed in the same synexpression group as fgf8, spry1, and spry2, all of which are associated with clefting in humans (thus bolstering the evidence for ism1 being a clefting gene).

15. The sentence (271-274) “Additionally, other members…..is accompanied by reduced expression of MMP9 and VEGFA” is unwieldy and should be more clearly written.

16. The sentence “A recent study has reported ISM1….at the serine residue 473” is unwieldy and unclear. It should be reworded and possibly split into two sentences.

Author Response

  1. In the abstract, lines 12 and 13, craniofacial development is included.
  2. Lines 33-26: “Ism” is replaced with “Ism1”.
  3. Line 39: “Mentioned” is replaced with “indicated.” This grammatical issue is reviewed throughout the paper.
  4. Line 58: The summary of Figure 1 is added and explained. Also, it is made clear that the phylogenetic analysis was based on proteins.
  5. In Figure 2 and in its legend, correction is done accordingly.
  6. The sequence of sentence is rearranged as instructed.
  7. For Figure 3, a and b for the two halves of the cartoon are added.
  8. Figure 4: The alignment is rearranged starting with ISM1 and then ISM2 followed by the other proteins.
  9. Figure 5: a or b for the two halves of the cartoon is added.
  10. Line 181: “ISM is expressed in several human tissues” is changed to “ISM1 is expressed in a large number of human tissues,” and additional examples are added.
  11. Lines 220-221: The statement is reviewed, and correction is made accordingly.
  12. Line 225: “craniofacial development” is included in the list of events.
  13. ISM1 has been shown to interact with integrins. A brief list of integrin functions is added to this review. Thus, it will be clear for readers to understand the role of ISM1 through integrin as in that specific part of review, example of endothelial cell apoptosis through integrin is also described for clarification.
  14. The Lansdon et al paper (2018) was recited in the reference section and a brief discussion of craniofacial defects were added in “Pathological processes”
  15. The sentence (271-274) is revised and clearly written.
  16. The sentence is broken down into two sentences for the clarification and further rewording is done for clear understanding of reader.

Reviewer 2 Report

In the present review, Shakhawat, H et al. reviews Isthmin protein. Authors provides a perspective on the gene organization and evolution of the ISM family, their links with developmental and physiological functions, and key questions for the future.

The review is exhaustive and presents detailed overview of Isthimin protein and should be published in Cells journal.

Author Response

N/A

Round 2

Reviewer 1 Report

Overall, the authors have done a satisfactory job addressing my concerns. My only suggestion would be to reword the last paragraph on page 11 to read something like the following (in order to be factually precise and put things in logical order):

Orofacial clefts have a complex etiology and are one of the most prominent birth defects, affecting 1–2 per 1000 births. Complex diseases have been shown to be associated with copy number variants (CNVs) and heterozygous deletions of ISM1 were found to be enriched in cleft lip and palate cases compared to controls thus demonstrating that the loss of even one copy predisposes to this disorder (Lansdon, Darbro, Petrin, Hulstrand, Standley, Brouillette, Long, Mansilla, Cornell, Murray, et al., 2018).  ISM1 is expressed in the same synexpression group as other clefting genes including spry1, spry2 and fgf8 and ISM1 knockdown causes craniofacial dysmorphologies in frogs including cleft-like phenotypes thereby confirming the role of ISM1 in craniofacial development (Lansdon, Darbro, Petrin, Hulstrand, Standley, Brouillette, Long, Mansilla, Cornell, Murray, et al., 2018).

Author Response

  1. Last paragraph on page 11 is reworded and arranged as per the reviewer suggestion.

Moreover, orofacial clefts have a complex etiology and are one of the prominent birth defects, affecting 1–2 per 1000 birth. Complex diseases have been reported to be linked with copy number variants (CNVs) and ISM1 heterozygous deletions were shown to be enriched in cleft lip and palate cases compared to the controls hence indicating that loss of even one copy predisposes to this disorder Lansdon, Darbro, Petrin, Hulstrand, Standley, Brouillette, Long, Mansilla, Cornell, Murray, et al., 2018). SM1 is expressed in the same synexpression group as other clefting genes including spry1, spry2 and fgf8 and knockdown of ISM1 causes craniofacial dysmorphologies in frogs including cleft-like phenotypes hence confirming the role of ISM1 in craniofacial development (Lansdon, Darbro, Petrin, Hulstrand, Standley, Brouillette, Long, Mansilla, Cornell, Murray, et al., 2018).